# Language Model Fine-Tuning on Scaled Survey Data for Predicting Distributions of Public Opinions

## Abstract

Large language models (LLMs) present novel opportunities in public opinion research by predicting survey responses in advance during the early stages of survey design. Prior methods steer LLMs via descriptions of subpopulations as LLMs' input prompt, yet such prompt engineering approaches have struggled to faithfully predict the distribution of survey responses from human subjects. In this work, we propose directly fine-tuning LLMs to predict response distributions by leveraging unique structural characteristics of survey data. To enable fine-tuning, we curate SubPOP, a significantly scaled dataset of 3,362 questions and 70K subpopulation-response pairs from well-established public opinion surveys. We show that fine-tuning on SubPOP greatly improves the match between LLM predictions and human responses across various subpopulations, reducing the LLM-human gap by up to 46% compared to baselines, and achieves strong generalization to unseen surveys and subpopulations. Our findings highlight the potential of survey-based fine-tuning to improve opinion prediction for diverse, real-world subpopulations and therefore enable more efficient survey designs.

## 1 Introduction

Surveys provide an essential tool for probing public opinions on societal issues, especially as opinions vary over time and across subpopulations. However, surveys are also costly, time-consuming, and require careful calibration to mitigate non-response and sampling biases (Choi & Pak, 2004; Bethlehem, 2010). Recent work suggests that large language models (LLMs) can assist public opinion studies by predicting survey responses across different subpopulations, explored in both social science (Argyle et al., 2023; Bail, 2024; Ashokkumar et al., 2024; Manning et al., 2024) and NLP (Santurkar et al., 2023; Chu et al., 2023; Moon et al., 2024; Hämäläinen et al., 2023; Chiang & Lee, 2023). Such capabilities could substantially enhance the survey development process, not as a replacement for human participants but as a tool for researchers to conduct pilot testing, identify subpopulations to over-sample, and test analysis pipelines prior to conducting the full survey (Rothschild et al., 2024).

Prior work in steering language models, *i.e.* conditioning models to reflect the opinions of a specific subpopulation, has primarily investigated different prompt engineering techniques (Santurkar et al., 2023; Moon et al., 2024; Park et al., 2024a). However, prompting alone has shown limited success in generating completions that accurately reflect the distributions of survey responses collected from human subjects. Off-the-shelf LLMs (Achiam et al., 2023; Dubey et al., 2024; Jiang et al., 2023) have shown to mirror the opinions of certain US subpopulations such as the wealthy and educated (Santurkar et al., 2023; Gallegos et al., 2024; Deshpande et al., 2023; Kim & Lee, 2023), while generating stereotypical or biased predictions of underrepresented groups (Cheng et al., 2023b;a; Wang et al., 2024). Furthermore, these models often fail to capture the variation of human opinions within a subpopulation (Kapania et al., 2024; Park et al., 2024b). While fine-tuning presents opportunities to address these limitations (Chu et al., 2023; He et al., 2024), existing methods fail to train models that accurately predict opinion distributions across diverse survey question topics and subpopulations.

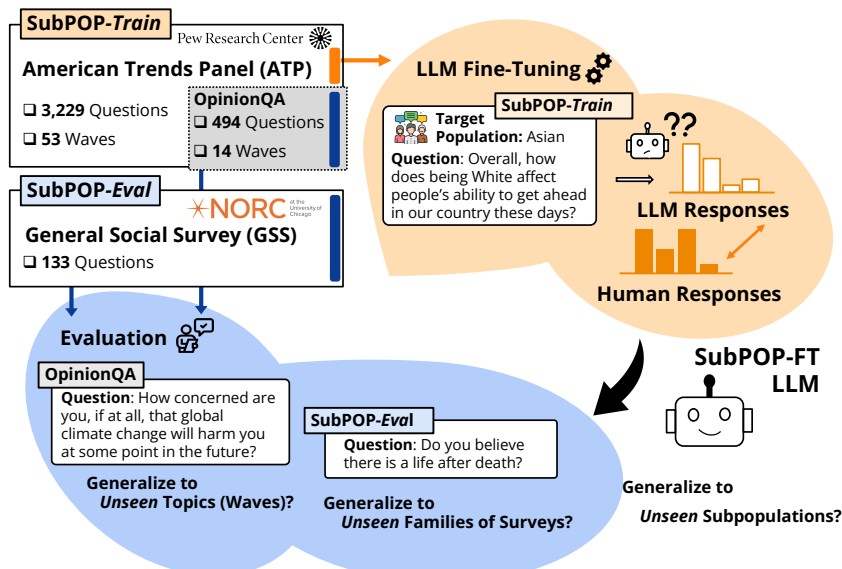

Figure 1: Illustration of our method and SubPOP. We collect survey data from two survey families—ATP from Pew Research (Pew Research Center, 2018) (forming SubPOP-Train) and GSS from NORC (Davern et al., 2024) (forming SubPOP-Eval). LLMs are fine-tuned on SubPOP-Train and evaluated on both OpinionQA (Santurkar et al., 2023) and SubPOP-Eval to assess generalization of distributional opinion prediction across unseen survey topics, survey families, and subpopulations.

**The present work.**  Here, we propose directly fine-tuning LLMs on large-scale, high-quality survey data, consisting of questions about diverse topics and responses from each subpopulation, defined by demographic, socioeconomic, and ideological traits. By casting pairs of (subpopulation, survey question) as input prompts, we train the LLM to align its response distribution against that of human subjects in a supervised manner. We posit that survey data is particularly well-suited for fine-tuning LLMs since: (1) We can train the model with clear **subpopulation-response pairs** that explicitly link group identities and expressed opinions, which is rare in LLMs' pre-training corpora, (2) Large-scale opinion polls are carefully designed and calibrated (*e.g.* using post-stratification) to estimate **representative** human responses, in contrast with LLMs' pre-training data where certain populations are over- or underrepresented, (3) Our training objective explicitly aligns model predictions with response **distributions** from each subpopulation, enabling LLMs to capture variance within human subpopulations.

Training on public opinion survey data has remained under-explored due to the limited availability of structured survey datasets. To this end, we curate and release SubPOP (**Sub**population-level **P**ublic **O**pinion **P**rediction), a dataset of 70K subpopulation-response distribution pairs (6.5× larger compared to previous datasets). We show that fine-tuning LLMs on SubPOP significantly improves the distributional match between LLM generated and human responses, and improvements are consistent across subpopulations of varying sizes. Additionally, the improvement generalizes to *unseen* subpopulations, survey waves (topics), and survey families, *i.e.* surveys administered by different institutions. Such broad generalization is particularly critical for real-world public opinions research, where practitioners are most in need of synthetic data for survey questions or subpopulations (or both) that they have not tested before.

Our contributions are summarized as follows:

- We show that training LLMs on response distributions from survey data significantly improves their ability to predict the opinions of subpopulations, reducing the Wasserstein distance between LLM and human distributions by 32-46% compared to top-performing baselines. (Section 4.1)

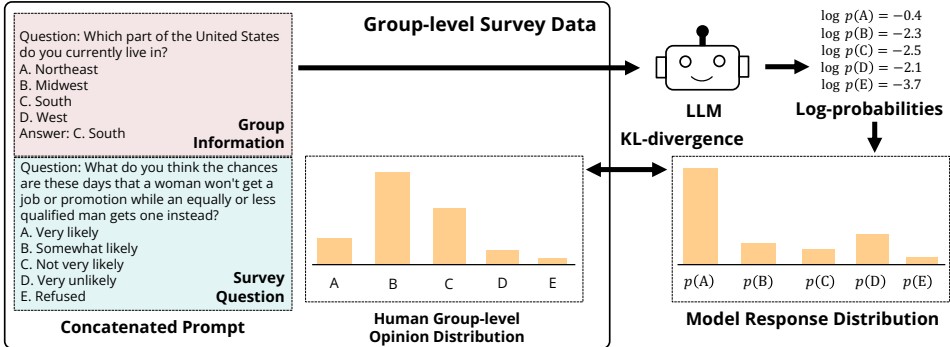

Figure 2: Proposed supervised fine-tuning setup with a survey response dataset such as SubPOP. Survey data is 3-tuple of a survey question, target subpopulation information, and the observed human opinion distribution (*i.e.* how subjects in the group responded to the given question). The training objective, $\mathcal{L}(\theta)$, is a forward KL divergence loss on language model predicted distribution of question option likelihoods; our loss guides the model predictions to match the response distribution of the specified human subpopulation.

- We show that the performance of the fine-tuned LLMs strongly generalizes to out-of-distribution data, including unseen subpopulations, new survey waves, and different survey families. (Section 4.1 and Section 4.2)
- We release SubPOP, a curated and pre-processed dataset of public opinion survey results that is $6.5\times$ larger than existing datasets, enabling fine-tuning at scale.

## 2 Related Work

**Public opinion datasets.** Several research institutions conduct large-scale public opinion polls and release data from those surveys. Important examples include Pew Research Center's American Trends Panel (ATP), which consists of multiple waves of cross-sectional surveys on different topics (Pew Research Center, 2018), and General Social Survey (GSS) from the NORC at the University of Chicago (Davern et al., 2024). Existing datasets have curated such data for evaluating LLM-based opinion predictions, including OpinionQA (Santurkar et al., 2023), a subset of ATP survey waves containing about 500 questions on contentious social topics. While OpinionQA is widely used in prior work (He et al., 2024; Zhao et al., 2023; Li et al., 2023; 2024), we find its total number of questions limited in scale for fine-tuning LLMs and instead use this dataset for evaluation. We further collect an extended set of survey data from ATP waves not included in OpinionQA, as well as from GSS to curate SubPOP. Other datasets, such as GlobalOpinionQA (Durmus et al., 2023)—derived from the World Values Survey (WVS) (World Values Survey, 2022) and the Pew Global Attitudes Survey (Pew Research Center, 2024)—and the PRISM dataset (Kirk et al., 2024) investigates how language models align with opinions from populations across the globe and cultures.

For additional discussions of related work, refer to Appendix C.

## 3 Methods

### 3.1 Fine-tuning LLMs on Human Response Distributions

Our goal is to fine-tune an LLM to predict the distribution of responses for a multiple-choice question, conditioned on descriptions of a human subpopulation we want to simulate, typically a specific demographic, socioeconomic, or ideological subgroup. Consider the example in Figure 2: the question asks, "What do you think the chances are these days that a woman won't get a job or promotion while an equally or less qualified man gets one instead?" The available responses are: *A. Very likely, B. Somewhat likely, C. Not very likely, D.*

104 *Very unlikely, and E. Refused.* In this case, the LLM will output a probability for each of the
105 tokens corresponding to the choices A through E, thereby generating a complete response
106 distribution that we aim to align with the true distribution observed in survey data.

107 Formally, let $q \in Q$ be a question, $g \in G$ be a subpopulation, and $\mathcal{A}_q$ be the set of possible
108 choices for question $q$. An LLM with parameters $\theta$ produces a conditional probability
109 distribution $p_\theta(\mathcal{A}_q \mid q, g)$. We fine-tune this model so that its predicted distribution for
110 each $(q, g)$ mirrors the human response distribution $p_H(\mathcal{A}_q \mid q, g)$ collected from real
111 survey data. To accomplish this, we use LoRA (Hu et al., 2021) and the forward Kullback–
112 Leibler (KL) divergence as our loss. Concretely, if $p_H(\mathcal{A}_q \mid q, g)$ represents the group-level
113 empirical distribution of human opinions and $p_\theta(\mathcal{A}_q \mid q, g)$ represents the model's predicted
114 distribution, our training objective is:

$$\mathcal{L}(\theta) = \mathbb{E}_{q,g}\Big[D_{\mathrm{KL}}\big(p_H(\mathcal{A}_q \mid q, g) \,\|\, p_\theta(\mathcal{A}_q \mid q, g)\big)\Big],$$

115 where $D_{\mathrm{KL}}$ denotes the KL divergence. In the example shown in Figure 2, the model is
116 trained to reduce the KL divergence between the target (survey-based) distribution over
117 $\{A, B, C, D, E\}$ and its predicted distribution for the subpopulation living in the Southern
118 United States. We choose forward KL (i.e., $\mathrm{KL}\big(p_H \,\|\, p_\theta\big)$) since it is sensitive to cases where
119 $p_H$ assigns high probability but $p_\theta$ does not, naturally encouraging the model to *cover* the
120 real distribution. This property aligns with standard maximum-likelihood training, where
121 the model is penalized for underestimating any response that is frequent in the data. In
122 other words, if many participants in group $g$ choose option "A" for question $q$, then the
123 model probability on "A" should be correspondingly high.

124 Instead of explicitly modeling the group response distribution as $p_H(\mathcal{A}_q|q, g)$, one could
125 do two alternatives. (1) One-hot encoding: this approach (Li et al., 2024) approximates
126 the distribution by a one-hot vector, assigning a value of one to the most probable option
127 and zero elsewhere. (2) Data augmentation by response frequency: this approach (Zhao
128 et al., 2023) expands the dataset by replicating question-choice pairs in proportion to their
129 observed frequency. We adopt the explicit distribution modeling in our main experiments
130 because it directly encodes the distributional information without requiring discrete sam-
131 pling or replicating data points. A detailed comparison of these approaches is provided in
132 Section F.2.

### 3.2 SubPOP: a Comprehensive Survey Dataset to Fine-tune and Evaluate LLMs

134 OpinionQA (Santurkar et al., 2023) is a widely used dataset for fine-tuning and evaluating
135 large language models (LLMs) on opinion prediction, containing roughly 500 questions
136 drawn from 14 American Trends Panel (ATP) waves (Pew Research Center, 2018). Although
137 valuable, it faces two important limitations: (1) Limited thematic diversity—for instance,
138 wave 26 focuses on the topic of firearms. (2) Reliance on a single survey family (ATP), which
139 risks overfitting to a particular style of questions and limits out-of-distribution evaluation
140 on other sources (e.g., GSS).

141 To address these limitations, we introduce a new dataset, SubPOP, that broadens both the
142 thematic and institutional scope of opinion prediction data. For training, SubPOP comprises
143 3,229 multiple-choice questions drawn from ATP waves 61–132, excluding waves included
144 in OpinionQA. In Table 4, we list the topics of the ATP waves in SubPOP vs. OpinionQA,
145 both showing the increased thematic diversity of SubPOP (with over 20 new topics) and the
146 remaining unseen topics in OpinionQA that allow us to test whether LLMs fine-tuned on
147 SubPOP can generalize to unseen topics.

148 For evaluation, SubPOP also includes 133 multiple-choice questions from the General Social
149 Survey (GSS) (Davern et al., 2024), serving as an out-of-distribution benchmark. This
150 expanded collection not only broadens the range of topics beyond OpinionQA's initial 500
151 questions, but also enables evaluation on surveys created and administered by different
152 institutions (Pew Research Center vs. NORC-Chicago). Dataset curation and refinement
153 pipeline is available in Appendix D.

Table 1: Evaluation on OpinionQA and the SubPOP evaluation set (SubPOP-Eval) for 22 subpopulations following Santurkar et al. (2023). We compute the WD by averaging over all questions and subpopulations. Lower and upper bounds of performance give guidance on how each method performs. For Modular Pluralism, we provide an error rate of one-hot prediction (†) (Section 3.3) which was used in the original paper.

| Method | OpinionQA | | | | SubPOP-Eval | | | |
|---|---|---|---|---|---|---|---|---|
| | Llama-2-7B | Llama-2-13B | Mistral-7B | Llama-3-70B | Llama-2-7B | Llama-2-13B | Mistral-7B | Llama-3-70B |
| Upper bound (Unif.) | 0.178 | | | | 0.208 | | | |
| Lower bound (Human) | 0.031 | | | | 0.033 | | | |
| Zero-shot prompt (QA) | 0.173 | 0.170 | 0.153 | 0.138 | 0.206 | 0.196 | 0.187 | 0.160 |
| Zero-shot prompt (BIO) | 0.193 | 0.183 | 0.162 | 0.143 | 0.221 | 0.212 | 0.202 | 0.175 |
| Zero-shot prompt (PORTRAY) | 0.195 | 0.207 | 0.158 | 0.209 | 0.212 | 0.242 | 0.194 | 0.247 |
| Few-shot prompt | 0.186 | 0.175 | 0.174 | 0.166 | 0.217 | 0.194 | 0.175 | 0.182 |
| Modular Pluralism | 0.285 (†55.6%) | | | | 0.279 (†55.2%) | | | |
| Ours (SubPOP-FT) | 0.106 | 0.102 | 0.096 | 0.094 | 0.121 | 0.113 | 0.115 | 0.096 |

## 3.3 Evaluation Metric

We use Wasserstein distance (WD) to quantify how closely the model's predicted opinion distribution matches human survey data (Santurkar et al., 2023; Moon et al., 2024; Meister et al., 2024; Zhao et al., 2023). Formally, for a group $g$ representing some subpopulation and a question $q$ WD is defined as $\mathcal{WD}_\theta(q,g) = \mathcal{WD}(p_H(\mathcal{A}_q|q,g), p_\theta(\mathcal{A}_q|q,g))$ (see formula in Appendix E). Since WD is computed over ordinal values, we map the categorical answer options to numbers, such as mapping "Very likely" to 1, "Likely" to 2, and so on.

Some prior work utilizes one-hot accuracy (Feng et al., 2024; Li et al., 2023) as an evaluation metric. However, one-hot accuracy only verifies whether the top-predicted choice matches the top human response, thereby discarding distributional information. In contrast, WD accounts for partial overlaps among the categories and reflects the 'cost' of shifting probability mass, providing a more nuanced assessment of distribution discrepancy. Consider the example question provided in Figure 2, where the human response distribution indicates that option B ("Somewhat likely") is the most probable. Now consider two cases in which the model incorrectly predicts the top choice. In the first case, the model assigns a high probability to option A ("Very likely"), while in the second case, it assigns a high probability to option D ("Very unlikely"). Although one-hot accuracy would treat both predictions equally as errors, WD differentiates between them by accounting for the ordinal relationship among the options, penalizing the second prediction more heavily for its larger deviation from the true distribution.

## 4 Experiments

**Lower and upper bounds.** We use a uniform distribution over all available choices to establish an upper bound of the WD between a predicted and the target response distribution. To compute a lower bound, we sample a group of human respondents from the original human respondents to calculate the WD between the two, and perform bootstrapping to obtain a robust estimate. This lower bound captures the intrinsic variance arising from the respondent sampling process in opinion surveys.

**Baselines.** We compare our approach with two baseline methods: prompting and Modular Pluralism (Feng et al., 2024). For prompting, we consider both zero-shot and few-shot methods. In zero-shot prompting, we steer the LLM using demographic prompt formats. Specifically, we employ three different formats following Santurkar et al. (2023): QA, BIO, and PORTRAY. For instance, to condition the LLM to a person living in the South of the US, the QA format uses a question-answer format as illustrated in Figure 2; the BIO format conditions the model with a first-person narrative such as "I currently reside in the South."; and the PORTRAY format uses a third-person narrative like "Answer the following question as if you currently reside in the South.".

Few-shot prompting augments the prompt with a few examples of question-response distribution pairs alongside the demographic label (Hwang et al., 2023). In particular,

we select the top five few-shot examples from the SubPOP training set based on cosine similarity computed by the embedding model. In our experiments, we represent the response distribution in JSON format and require the model to output its prediction in the same JSON format, following the approach in Meister et al. (2024).

Modular pluralism (Feng et al., 2024) fine-tunes multiple LLMs on distinct datasets to capture the viewpoints of different communities (Feng et al., 2023). For a given question, each fine-tuned LLM generates an opinion that reflects the perspective of the community it represents, and a separate black-box LLM aggregates these outputs to produce the final distributional response. Detailed implementation of the lower/upper bounds and the baselines is provided in Appendix G.

## 4.1 Generalization to Unseen Topics and Survey Families

In this section, we assess the ability of our fine-tuned LLMs to generalize to unseen data—both in terms of new topics and entirely different survey families. To evaluate these aspects, we use OpinionQA to measure generalization to unseen topics, and SubPOP-Eval to test generalization to a different survey family. We fine-tune four LLMs (Llama-2-7B, Llama-2-13B, Mistral-7B, and Llama-3-70B) on SubPOP-Train. We opt for pretrained LLMs rather than instruction-following models, as previous work has shown that pretrained models perform better on this task (Moon et al., 2024). A detailed comparison between these model types is provided in Appendix F.3.

**Summary of results.** Table 1 reports the average WD metrics computed over all demographic groups and survey questions, comparing our fine-tuned models against various baseline approaches. Our experiments show that fine-tuning on SubPOP-Train significantly outperforms all other methods, yielding a 32–46% reduction in WD on OpinionQA and a 39–42% reduction on SubPOP-Eval compared to the best baselines. Notably, SubPOP-Train is based on ATP data, while SubPOP-Eval is derived from GSS surveys—two distinct survey families that can differ in respondent pools, calibration techniques, and other methodological factors, leading to non-trivial distribution shifts despite both being representative of the US population. Furthermore, our fine-grained analyses at the wave level (see Appendix H) confirm that these trends persist even at more detailed levels of evaluation.

**Comparison to zero- and few-shot prompting.** We first compare the performance of prompting methods with our approach. Zero-shot prompting results in only modest WD improvements over the upper bound, with the largest gain observed for Llama-3-70B and negligible improvements for Llama-2-7B. Even when using few-shot prompting—where five example question-response distribution pairs are provided—the performance gains remain minimal. This may be partly due to an under-optimized prompt format (*e.g.* requiring JSON output) and the inherent sensitivity of language models to prompt formatting (Sclar et al., 2023; Anagnostidis & Bulian, 2024). These findings underscore the need for methods, such as fine-tuning, that enable relatively reliable predictions of opinion distributions.

**Comparison to Modular Pluralism.** Modular Pluralism improves one-hot accuracy, reducing prediction error from 72.7% (zero-shot prompting) to 55.6% on OpinionQA, but underperforms in matching the full distribution of option choices, measured as WD. This discrepancy in performance highlights the limitations of methods that train LLMs to identify only the most probable response rather than modeling the entire distribution of responses. Opinions are inherently distributed: even within a particular subpopulation such as a single demographic subgroup, distribution of opinions cannot be captured as a single most likely response. Moreover, instruction-tuned models that serve as a black-box LLM tend to assign high probabilities on only specific tokens (Lin et al., 2022; Kadavath et al., 2022; Achiam et al., 2023), further pushing the generated distribution away from the human distribution.

## 4.2 Generalization across Target Subpopulations

Here we report two key observations: (1) prediction performance improves consistently across subpopulations in the fine-tuning data, and (2) the LLMs fine-tuned on SubPOP-Train generalize well to subpopulations that were not included during fine-tuning.

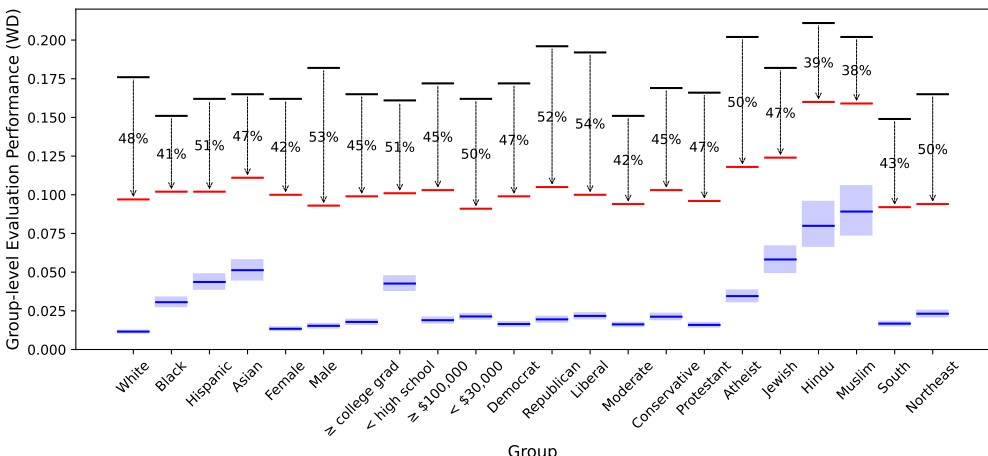

Figure 3: Per-group evaluation performance of our model Llama-2-7B-SubPOP-FT (red lines) on OpinionQA. For comparison, the results from zero-shot QA prompting (black lines) and the lower bound (blue lines) are presented. We observe that the relative improvement, measuring how much of the gap between zero-shot prompting and the lower bound has been closed, remains consistent across subpopulations. Shaded blue regions represent the 95% confidence interval of the lower-bound estimation for each group. Per-group results for other models (Table 9) and the results on SubPOP evaluation set (Table 10) are available in Appendix H.

**Consistent performance improvements over subpopulations.** Figure 3 shows the per-group WD on the OpinionQA evaluation for Llama-2-7B, comparing our fine-tuning approach with zero-shot prompting and the empirical WD lower bound. To evaluate the consistency of performance gains, we calculate the *relative improvement* for each subpopulation as how much of the gap between zero-shot prompting and the empirical lower bound is reduced after fine-tuning. This measure allows us to account for varying lower bounds across subpopulations: since some groups have fewer respondents, there is greater uncertainty in their reported distribution in the survey data and greater variance between the original sample and bootstrap samples.

All 22 subpopulations demonstrate a large relative improvement after fine-tuning, ranging from 38%–54%. The average relative improvement is 46.7% with a standard deviation of 4.4%. This consistency confirms that our fine-tuning approach delivers balanced performance gains without disproportionately favoring any particular demographic subgroup. We hypothesize that the consistent gains over groups largely stem from our dataset design, which allocates an equal number of training samples to each group. By ensuring uniformly distributed data points across subpopulations, the model captures sufficient subgroup-specific signals, ultimately leading to consistent performance improvements.

**Generalization to unseen subpopulations.** We further investigate how models fine-tuned with our approach and SubPOP might show generalization to subpopulations that were not represented in the training data, a circumstance that may arise in real-world survey development. For the evaluation, we benchmark our methods against a zero-shot prompting baseline. Specifically, we evaluate our model, which is fine-tuned on 22 subpopulations provided in SubPOP-Train, on a set of subpopulations in OpinionQA that were not included in fine-tuning. This experiment not only checks generalization to unseen subpopulations, but also involves unseen survey questions, providing a robust assessment of the model capability for generalization to out-of-distribution data.

As shown in Table 2, our model achieves a strong reduction in WD even for unseen subpopulations, indicating that the model can be steered by demographic prompts beyond the seen subpopulations in training. Interestingly, although SubPOP-Train does not contain any data with opinion distributions of particular age groups (*e.g.* subjects of age 18-29 or those of age 65+), the average relative improvement is 44.7%, which is compatible with the

Table 2: Per-group evaluation performance of Llama-2-7B-SubPOP-FT (Ours) on OpinionQA. We report the lower bound, WD for zero-shot prompting, WD for Llama-2-7B-SubPOP-FT, and the relative improvement. Rows highlighted in blue represent subpopulations included during fine-tuning, while uncolored rows correspond to subpopulations that were unseen during fine-tuning.

| Group | Lower Bound | Zero Shot | Ours | Relative Improvement (%) |
|---|---|---|---|---|
| Age: 18-29 | 0.023 | 0.185 | 0.096 | 54.9 |
| Age: 30-49 | 0.014 | 0.151 | 0.093 | 42.3 |
| Age: 50-64 | 0.014 | 0.154 | 0.101 | 37.9 |
| Age: 65+ | 0.013 | 0.195 | 0.115 | 44.0 |
| Less than high school | 0.043 | 0.161 | 0.101 | 50.8 |
| High school graduate | 0.017 | 0.144 | 0.092 | 40.9 |
| Some college, no degree | 0.018 | 0.144 | 0.093 | 40.5 |
| Associate's degree | 0.026 | 0.159 | 0.098 | 44.9 |
| College grad | 0.018 | 0.165 | 0.099 | 44.9 |
| Postgraduate | 0.015 | 0.174 | 0.106 | 42.8 |
| Very conservative | 0.026 | 0.208 | 0.107 | 55.5 |
| Conservative | 0.021 | 0.169 | 0.103 | 44.6 |
| Moderate | 0.016 | 0.151 | 0.094 | 42.2 |
| Liberal | 0.022 | 0.192 | 0.100 | 54.1 |
| Very liberal | 0.025 | 0.202 | 0.111 | 51.4 |
| Democrat | 0.016 | 0.172 | 0.099 | 47.1 |
| Republican | 0.019 | 0.196 | 0.105 | 52.0 |
| Independent | 0.016 | 0.155 | 0.093 | 44.5 |
| Something Else | 0.026 | 0.162 | 0.092 | 51.0 |

average relative improvement for seen subpopulations. We provide results for other unseen groups in Table 7 of Appendix F.4 (average relative improvement of 43.1% with a standard deviation of 6.7%).

**Steerability towards subpopulations.** Given the large improvements in WD across subpopulations after fine-tuning, we want to test whether the LLM is truly adapting its predictions based on the subpopulation specified in its prompt (*i.e.* the LLM is being steered) or if the improvements can be explained by the LLMs' predictions getting closer to human responses in general, without any subpopulation-specific adaptation. If the LLM is being steered, we should expect that the LLM's predictions for a target subpopulation $g_t$ are closer to the human distribution for $g_t$ when $g_t$ is the subpopulation specified in the prompt, compared to when another group $g_s$ is specified in the prompt. We should also expect the gap in WD to be larger if the distance between the true human distributions for $g_t$ and $g_s$ are larger, such as differences between the youngest and oldest age groups compared to adjacent groups.

Formally, we define the *intergroup disagreement* between a target group $g_t$ and a source group $g_s$ as $\mathcal{WD}(p_H(\mathcal{A}_q \mid q, g_t), p_H(\mathcal{A}_q \mid q, g_s))$ averaged over evaluation questions. In human responses (left of Figure 4), the disagreement shows the pattern of locality: increases as the disparity in education levels between two groups grows. We extend this notion to compare the human distribution from the target group $g_t$ with the LLM-predicted distribution when the *source* group $g_s$ is specified in the prompt, $\mathcal{WD}(p_H(\mathcal{A}_q \mid q, g_t), p_\theta(\mathcal{A}_q \mid q, g_s))$. If the model truly incorporates subpopulation information from the prompt, its intergroup disagreement pattern should mirror that of the human data.

Zero-shot prompting with the base model (right of Figure 4) does not exhibit the locality pattern seen in the human data, indicating that it cannot be steered by subpopulation labels. In contrast, the fine-tuned model (middle of Figure 4) reproduces a pattern resembling the human-human case, even though it was trained on only two education groups ("less than high school" and "college graduate/some postgrad") and the other four groups were unseen. This result demonstrates that our fine-tuned model not only learns to condition

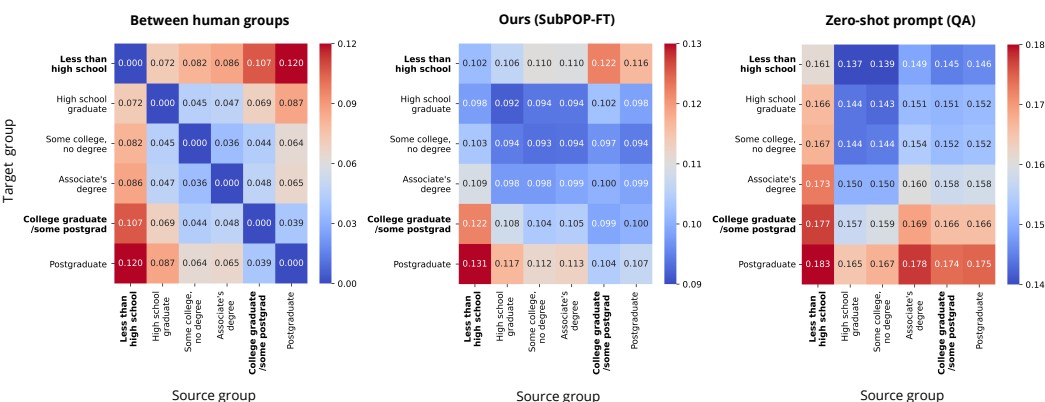

Figure 4: *Intergroup disagreement* pattern between groups of different education levels calculated with OpinionQA and Llama-2-7B as a base model. A target human group is compared to (left) a source human group, (middle) our fine-tuned model conditioned on a source group, (right) a base model conditioned on a source group. Bold-faced groups are included in the fine-tuning data SubPOP-Train, while the others aren't. In the human response (left), we observe a decreasing disagreement level as the education level becomes similar. This disagreement pattern exists in our fine-tuned model but not in the zero-shot prompting with a base model, indicating that our model can be steered to given subpopulation label even for unseen demographics while the base model cannot.

on subpopulation information but also generalizes to subpopulations unseen during fine-tuning. We provide the intergroup disagreement for other traits in Appendix F.4.

## 5  Conclusion

In this work, we demonstrated that fine-tuning large language models on structured public opinion survey data markedly improves their ability to predict human response distributions. We curate SubPOP —a dataset 6.5× larger than previous collections to fine-tune and evaluate LLMs on survey response distribution prediction. By fine-tuning on SubPOP, we showed that LLMs can capture the group-specific variability in public opinions, also generalizing to unseen subpopulations, survey waves and question topics, and different survey families. Fine-tuning achieves consistent improvements across subpopulations of varying sizes, and our experiments demonstrate that fine-tuned LLMs are indeed *adapting* their responses to the subpopulation specified in the prompt, even for subpopulations unseen during fine-tuning. Finally, our experiments also reveal that as the fine-tuning dataset grows, model performance continues to scale favorably, underscoring the value of our larger dataset.

Generalization is a critical capability for LLMs if they are to be used to assist public opinion research, as researchers are most in need of opinion predictions for questions or subpopulations whom they have not surveyed before. Our work, by greatly improving LLMs' ability to accurately predict opinions with fine-tuning and demonstrating strong generalization to out-of-distribution data, moves us closer towards the goal of leveraging LLMs for opinion prediction. However, many open questions remain: why is the model able to generalize well to unseen subpopulations and questions, and when might it fail to do so? How do we ensure that LLMs capture opinions along other dimensions not explored in this work, such as intersections of demographic identities or temporal change? How should LLMs be integrated into survey designs, to serve as tools that can complement surveys with human participants? Answering these questions will require interdisciplinary collaborations with domain experts and critical assessments of LLMs' and traditional survey methods' strengths and weaknesses.

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

## A    Limitations

In this work, we explore the capability of language models to complement traditional survey design by predicting survey responses in advance. However, we acknowledge the following inherent limitations of this approach.

**Role in Survey Research.**    While language models can provide a coarse approximation of human opinions, they cannot fully replace human involvement in the survey process. Human opinions evolve dynamically in response to social events, and while pretrained language models can incorporate such knowledge through retrieval-augmented generation, they remain limited in adapting to a rapidly changing world. Moreover, fine-tuning a language model on distributions of human opinions may inadvertently replicate and amplify existing biases of humans, leading to undesirable outcomes. It is important to note that a model fine-tuned on human opinions does not necessarily align with human values and behaviors, nor does it serve as a perfect proxy for human decision-making. The scope of our work is restricted to language models prompted with a group-level information generating response distributions to survey questions, rather than simulating individual human respondents in a personalized manner.

**Data Dependence.**    Survey response data, even after post-stratification calibration, remain subject to empirical variance, particularly for relatively small groups that comprise about one percent of the U.S. population. Also, while traditional surveys have implemented various strategies to mitigate response bias stemming from the linguistic and multiple-choice nature of survey questions (Tourangeau, 2000), the extent to which these biases affect language models—and how best to address them—remains an open question (Tjuatja et al., 2024; Bisbee et al., 2024). Future research could focus on developing reliable opinion datasets for underrepresented groups and examining how prompt engineering elements can be optimized to reduce bias in language model-generated responses.

**Limited Contextual Information.**    Our fine-tuning approach, which structures prompts in a QA format, demonstrates strong matching with human opinion distributions. However, we have not explored fine-tuning with richer contextual information. Prior research suggests that incorporating additional contextual details can improve the fidelity of model-generated opinions to actual human responses. We anticipate that more sophisticated steering techniques could further enhance the opinion prediction performance beyond the results presented in this study. Investigating such methods remains an open and promising direction for future work.

## B    Potential Risks

Employing language models for opinion prediction has both influential possibilities and risk of misuse. We acknowledge that the risk of misuse cannot be overlooked, and we clearly state that indiscriminately minimizing the discrepancy of opinion response distribution as a fine-tuning target can cause severe harms. In particular, the model might develop a bias toward specific demographics during the course of fine-tuning, an artifact of minimizing response distribution when other safeguard measures are not employed. We emphasize that an oversight and holistic evaluation of methods and pipelines are required before deploying such models for any of the actual applications and interactions with human.

## C    Expanded Discussion of Related Work

**Predicting human opinions with LLMs.**    Prior work has explored various prompt engineering approaches for steering LLM responses: earlier work use rule-based prompts that incorporate demographic profiles of individuals or populations, or few-shot examples of survey question-response (Hwang et al., 2023; Simmons, 2022; Santurkar et al., 2023; Dominguez-Olmedo et al., 2023). Recent work explore prompting LLMs with open-ended text, including interview transcripts (Park et al., 2024a), personal narratives (Moon et al.,

2024), or LLM-refined prompts (Kim & Yang, 2024; Sun et al., 2024). Our fine-tuning approach is complementary to prompt engineering methods: while prompt engineering seeks to optimize what information is provided to the LLM (while the model is frozen), fine-tuning seeks to optimize how the model utilizes the provided information (while the prompt is frozen). In this work, we demonstrate that our fine-tuned models exhibit significant improvements in matching the response distributions of humans without requiring elaborate prompt engineering methods.

Other work (Chu et al., 2023; He et al., 2024; Feng et al., 2024) fine-tune language models on text corpora from specific communities (*e.g.*, Reddit) to infer the most popular response or response distribution for a given survey question. While this approach benefits from large-scale and continuously updated text corpora, it struggles with disproportionate representation online and lacks comprehensive coverage of diverse subpopulations. A few works have explored directly fine-tuning on public opinion survey data, but in different problem settings from ours. Li et al. (2023) apply collaborative filtering to individual-level responses to learn embeddings for individuals, and Zhao et al. (2023) develop a meta-learning framework to predict the opinions of new groups given a small number of in-context examples for that group. In contrast, our approach does not require individual-level responses and can generalize to unseen groups and survey questions without *any* responses.

A recent work (Li et al., 2024) and a work concurrent to ours (Cao et al., 2025) also explored fine-tuning LLMs on the World Values Survey (WVS) to align the LLM's opinion response with a culture or entire country populations. In comparison, our work focuses on US surveys, testing whether LLMs can align with finer-grained subpopulations within one country and whether LLMs fine-tuned on one US-representative survey can generalize to another. However, we note that our proposed method for fine-tuning language models applies to any survey dataset with distributional information about subpopulation responses.

**Pluralistic alignment of LLMs.**   Recent literature on pluralistic and distributional alignment target a similar yet different problem in fine-tuning LLMs (Chakraborty et al., 2024; Melnyk et al., 2024; Poddar et al., 2024; Siththaranjan et al., 2023; Yao et al., 2024; Sorensen et al., 2024; Lake et al., 2024; Chen et al., 2024; Jiang et al., 2024). While this line of work shares a similar goal as ours in training models to reflect on opinions (and preferences) of diverse subpopulations, most work differ from ours in that they operate in the context of training against *pair-wise* preference orderings between alternative language model completions, extending the Bradley-Terry-Luce model (Rajkumar & Agarwal, 2014; Ouyang et al., 2022; Rafailov et al., 2024) or investigating alternative models to account for diverging preference orderings across populations. In contrast, our work trains the model to directly predict the opinion distributions of human subpopulations, where accurately matching distributions across a large variety of subpopulations is of paramount interest. Our work additionally focuses on the particular context of estimating human opinions about societal issues—the objective of public opinion research—which enables relatively straightforward supervised training on openly available, structured survey data as presented by SubPOP.

## D   Dataset Details

### D.1   American Trends Panel Datasets

Pew Research holds regular American Trends Panel (ATP) survey (called waves) (Pew Research Center, 2018) covering various topics (*e.g.* veterans, political priorities, gender and leadership) and releases result at an individual level. For each anonymized individual, the following information is released: unique identification number, demographic details, survey responses, and weight. Weights (Mercer et al., 2018) are the output of post-survey calibration process that helps adjusting survey results for response bias (e.g., non-response bias, sampling bias) correction and population representativeness. As of January 2025, survey data until wave 132 has been released. About 20 surveys are conducted in each year.

### D.2 OpinionQA

OpinionQA is a subset of ATP curated in (Santurkar et al., 2023). This dataset consists of contentious 500 questions sampled from 14 ATP waves which have high intergroup disagreement (i.e. large Wasserstein distances among subpopulations' responses to a question). It also comes with hand-crafted ordinality information which provides structure to option lists. For example, options 'Major reason', 'Minor reason', and 'Not a reason', are assigned an ordinality mapping to 1, 2, and 3, respectively. This ordinality allows a calculation of 1-dimensional Wasserstein distance.

Subpopulations we employ are listed in Table 3. This set of groups is adopted for several small-scale analysis (Santurkar et al., 2023; Zhao et al., 2023; Kim & Yang, 2024). We note that our approach is not limited to a specific number of groups and data is available for small or fine-grained demographic subpopulations.

Table 3: A list of 22 subpopulations used throughout our fine-tuning and analysis. We provide the number of respondents in each subpopulation in American Trends Panel Wave 82 for reference.

| Trait | Groups | Population % in Wave 82 |
|---|---|---|
| Region | Northeast | 17.2 |
| | South | 37.8 |
| Education | College grad+ | 24.2 |
| | Less than high school | 5.2 |
| Gender | Male | 44.3 |
| | Female | 54.6 |
| Race / ethnicity | Black | 9.6 |
| | White | 66.1 |
| | Asian | 4.8 |
| | Hispanic | 15.2 |
| Income | $100,000 or more | 21.8 |
| | Less than $30,000 | 21.3 |
| Political Party | Democrat | 35.1 |
| | Republican | 29.1 |
| Political Ideology | Liberal | 20.0 |
| | Conservative | 22.6 |
| | Moderate | 38.3 |
| Religion | Protestant | 40.8 |
| | Jewish | 2.0 |
| | Hindu | 0.9 |
| | Atheist | 0.6 |
| | Muslim | 0.7 |

### D.3 SubPOP-Train

We gather additional data from the American Trends Panel, specifically collecting 53 waves from Wave 61 to 132. There are 62 waves from Wave 61 - 132, however, some waves have missing demographic or ideology information (for example, wave 63 does not contain political ideology information) or the data is not available hence removed during the curation process. To refine the dataset, we exclude questions that meet the following criteria: those with more than 10 response options, redacted response data, or dependencies on prior questions (e.g., assessing political strength). For the remaining questions, we use GPT-4o to refine their wording, ensuring they are well-suited for prompting the language models while making minimal modifications. In Figure 5 we provide a few-shot prompt for question refinement.

In Figure 6, we visualize the embeddings of the question texts (projected to 2-dimensions using t-SNE) from OpinionQA compared to SubPOP-Train and SubPOP-Eval. The visualization shows how much larger our dataset is than OpinionQA (6.5×), along with the expanded coverage of our dataset into semantic areas untouched by OpinionQA. The embeddings also reveal the distribution shift from ATP questions to GSS questions: while the ATP and GSS questions overlap in embedding space, the GSS question appear as small clusters, not evenly

Figure 5: Few-shot prompt for refining the question to suit a language model prompting. An instruction is designed to make a minimal change to the original question, and in-context examples are provided.

distributed over the ATP questions. In Table 4, we list each ATP wave in SubPOP-Train and OpinionQA, along with its number of questions and wave topic(s), as defined by ATP.[1] The table indicates which topics are new in SubPOP-Train compared to OpinionQA, indicating the expanded coverage of our dataset, along with which topics remain unseen in OpinionQA, which we can use to test LLMs fine-tuned on SubPOP-Train for generalization.

## D.4 SubPOP-Eval

To further evaluate the out-of-distribution generalization ability of our fine-tuned models, we subsample 133 questions from the GSS 2022 dataset (Davern et al., 2024). We apply the same selection criteria as outlined in Appendix D.3, excluding questions that are redacted,

---

[1]ATP wave topics and time periods are defined at https://www.pewresearch.org/american-trends-panel-datasets/.

t-SNE of question text embeddings

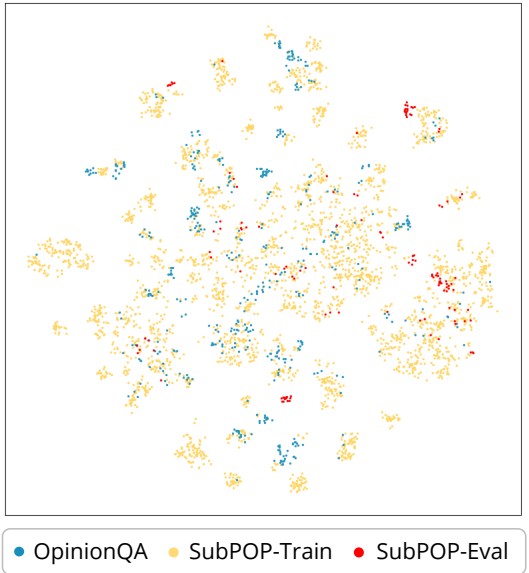

Figure 6: Embeddings of questions from OpinionQA, SubPOP-Train, and SubPOP-Eval.

conditioned on prior questions, inferable directly from the group information, derived from a set of questions, or those with more than 10 response options.

### D.5 Inspection of Identical Questions

Distribution of cosine similarities between two text embeddings (an output of the embedding model OpenAI-text-embedding-3-large given a question text), one from a question in SubPOP-Train and another from OpinionQA is shown in Figure 7. We observed a fraction of pairs having high cosine similarity, and manually inspected question pairs with high relevance. We find that by setting a threshold cosine similarity of 0.87 we can detect all semantically identical pairs. We took a conservative threshold of cosine similarity; this value was to maximize the recall at a cost of precision to ensure detection of overlapping questions.

## E   Experiment Details

We conduct our experiments using Nvidia A100 GPUs with 80GB VRAM. Hyperparameter tuning is performed over learning rates {5e-5, 1e-4, 2e-4} and batch sizes {64, 128, 256}. After evaluating possible combinations, we select a (learning rate, batch size) = (2e-4, 256) for Llama-2-7B, (learning rate, batch size) = (2e-4, 256) for Mistral-7B-v0.1, and (learning rate, batch size) = (1e-4, 256) for Llama-2-13B when utilizing the full training dataset. For Llama-3-70B, we have not done hyperparameter search but heuristically used (learning rate, batch size) = (2e-5, 256). For sub-sampled training data (Figure 9), we use the following configurations:

- (lr, bs) = (2e-4, 256) for 75% of the training data
- (lr, bs) = (1e-4, 128) for 50% of the training data
- (lr, bs) = (1e-4, 128) for 25% of the training data

All training is performed using LoRA (Hu et al., 2021), with LoRA parameters initialized from a normal distribution with $\sigma = 0.02$. We set the LoRA rank to 8, alpha to 32, and apply a dropout rate of 0.05. LoRA weights are applied to the query and value matrices. The AdamW (Loshchilov, 2017) optimizer is used with a weight decay of 0.

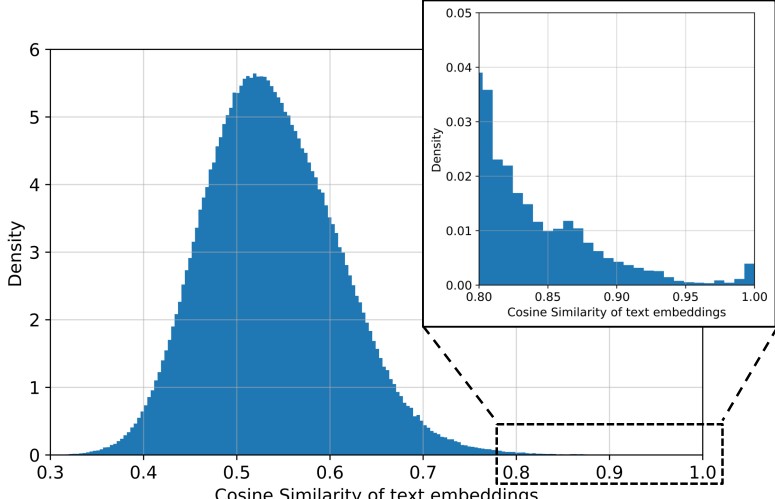

Figure 7: Distribution of cosine similarities between a question in SubPOP-ATP and Opin-ionQA, having a long tail towards a high cosine similarity. We inspect the question pairs in the range of 0.8 to 1.0 (distribution shown in the magnified view) and use a similarity of 0.87 as a safe threshold to identify a semantically identical question pair.

We use offline batched inference of vLLM (version 0.7.2) (Kwon et al., 2023) for inference and measuring response probability distribution of all methods.

**Choice of the Training Objective.** In this section, we explore both forward KL-divergence and Wasserstein Distance (WD) as training objectives. The forward KL-divergence is defined as

$$D_{\mathrm{KL}}(p_H \| p_\theta) = \sum_{a \in \mathcal{A}_q} p_H(a) \log \frac{p_H(a)}{p_\theta(a)},$$

where $p_H(a) \equiv p_H(a \mid q, g)$ and $p_\theta(a) \equiv p_\theta(a \mid q, g)$. Similarly, WD is given by

$$\mathcal{WD}(p_H, p_\theta) = \min_{\gamma \in \Pi(p_H, p_\theta)} \sum_{a, a' \in \mathcal{A}_q} \gamma(a, a') \, d(a, a'),$$

with $\Pi(p_H, p_\theta)$ denoting the set of all couplings between $p_H$ and $p_\theta$, and $d(a, a')$ the L1 distance between choices. Since survey responses are inherently one-dimensional and ordinal, we can simplify the computation of WD using cumulative distribution functions (CDFs). In the 1-D case, WD is computed as

$$\mathcal{WD}(p_H, p_\theta) = \int_{-\infty}^{+\infty} \left| F_{p_H}(x) - F_{p_\theta}(x) \right| \, dx,$$
$$= \sum_{i=1}^{n} \left| F_{p_H}(i) - F_{p_\theta}(i) \right|$$

where $F_{p_H}$ and $F_{p_\theta}$ are the CDFs corresponding to $p_H$ and $p_\theta$, respectively. We use this discrete formulation as the WD loss in our training.

While training with WD resulted in a higher KL-divergence on the validation set, the valida-tion WD converged to similar levels regardless of the objective (see Figure 8). We attribute this to KL-divergence penalizing low-probability assignments without significantly altering the overall distribution geometry. Given the KL divergence's broader applicability—without requiring ordinal information—we primarily used KL-divergence in our experiments.

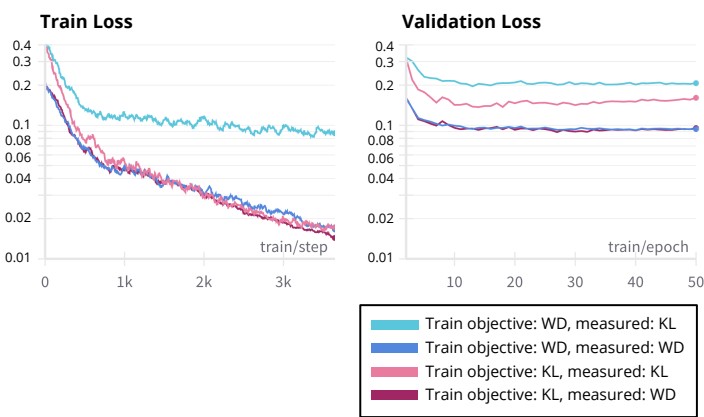

Figure 8: Train loss curve (left) and validation loss curve (right) for Llama-2-7B fine-tuned on 90% of OpinionQA, with the remaining 10% used for validation. Light and dark blue lines represent KL-divergence (KL) and Wasserstein distance (WD) when used KL as a training objective, while light and dark red lines represent KL and WD when used WD as a training objective. The two training objectives yield similar results in terms of WD, the primary measure of opinion distribution matching in our work.

# F  Additional Experiments

## F.1  Effect of Scaling the Dataset

In this section, we examine performance scales with training dataset size. We randomly sample subsets containing 25%, 50%, 75%, and 87.5% of the full SubPOP training set and evaluate three models—Llama-2-7B, Llama-2-13B, and Mistral-7B—on OpinionQA. As shown in Figure 9, we observe diminishing marginal returns, as is typical with fine-tuning; for example, after training on a random 25%, the models reach 72%-78% of the total improvement they achieve after fine-tuning on all of SubPOP-train. However, the performance does not entirely plateau. Instead, it continues to improve as we further increase the training data from 25% to 100%. We fit linear trend lines (dotted in Figure 9) to the results and observe that the slopes are similar for each model. This suggests that the rate of improvement—reflected by the slope in the power-law relationship—is intrinsic to the data and task rather than to the specific model architecture. In other words, LLMs exhibit comparable data efficiency, with performance gains that are fundamentally tied to dataset size rather than model-specific factors.

Using these trend lines, we can estimate the amount of fine-tuning data required to reach a target performance. For instance, we estimate that fine-tuning Mistral-7B on a dataset 25 times larger than the current SubPOP training set would yield a WD value of 0.07, which is much closer to the empirical lower bound of 0.031 reported in Table 1. This result underscores the critical importance of collecting more high-quality data, as increased dataset size can drive significant improvements in model performance.

## F.2  Effect of Response Distribution Modeling

In this section, we compare different methods for capturing the distribution of human responses. We consider three approaches:

1. *One-hot*: Predicting only the most probable response, which ignores the full distribution over all responses Li et al. (2024).

2. *Augment by N*: Augmenting the dataset by replicating each response by a factor of N according to its observed frequency Zhao et al. (2023).

3. *Explicit probability modeling*: Directly modeling the full response distribution using the actual probability values for each option.

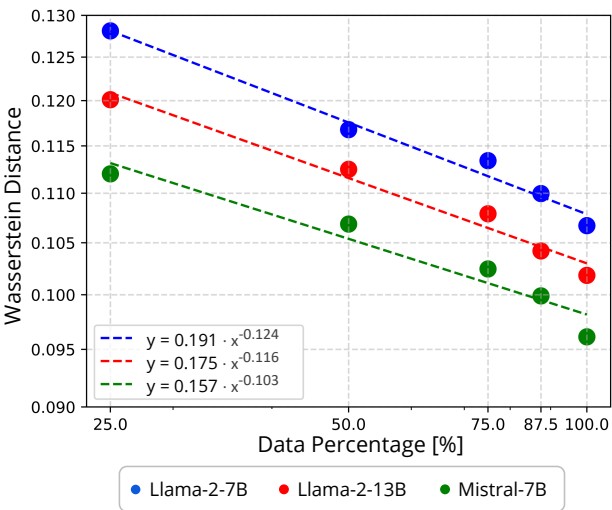

Figure 9: Evaluation results on OpinionQA after fine-tuning each LLM on increasingly large sampled subsets of `SubPOP-Train`. Both axes are presented in a log scale. The $x$-axis is the size of sampled dataset and the $y$-axis is WD against human responses measured on OpinionQA. Dashed lines represent a line of best fit. Performances at data percentage of 100% are identical to ours (`SubPOP-FT`) in Table 1.

Table 5 summarizes the results of these approaches. Notably, explicit probability modeling substantially outperforms the one-hot method, demonstrating that simply predicting the single most frequent response fails to capture the opinion diversity present within each subpopulation.

Compared with augment by $N$ (2nd and 3rd column in Table 5), explicit probability modeling also achieves better performance. Importantly, the performance gap exceeds the quantization error introduced by discretizing the response distribution. For instance, when discretizing with a factor of $N$, the quantization error is $\frac{1}{2N}$—approximately 0.01 or 0.005 in the cases shown in Table 5. Moreover, explicit modeling offers the practical benefit of reducing the data volume by a factor of $N$ compared to the augmentation approach, thereby lowering the computational cost of fine-tuning LLMs.

These results underscore the importance of explicit distribution modeling. By aligning the model's predictive distribution directly with the survey distribution, we achieve higher accuracy with fewer data samples, avoiding the rounding errors and replication overheads that are inherent to data-augmentation approaches.

### F.3 Post-trained Model

We fine-tune Llama-2-7B-chat to observe the effect of starting from checkpoints that have been instruction-tuned via Reinforcement Learning from Human Feedback (RLHF). Table 6 shows the evaluation performance of a baseline method (Zero-shot prompt (QA)), fine-tuned base model (Llama-2-7B) and fine-tuned chat model (Llama-2-7B-chat). We observe the significant performance improvement, while the baseline method performs worse then the models not instruction-tuned (Table 1). Especially, the performance for `SubPOP-Eval` of chat model is significantly worse than that of base model. We observe the high WD of the baseline method resulting from the model assigning high probability to a specific token (e.g. 'A'), being far apart from the human opinion distribution. After fine-tuning the model are able to generate a more distributed probability of answer tokens. This result coincides with the result reported in Moon et al. (2024).

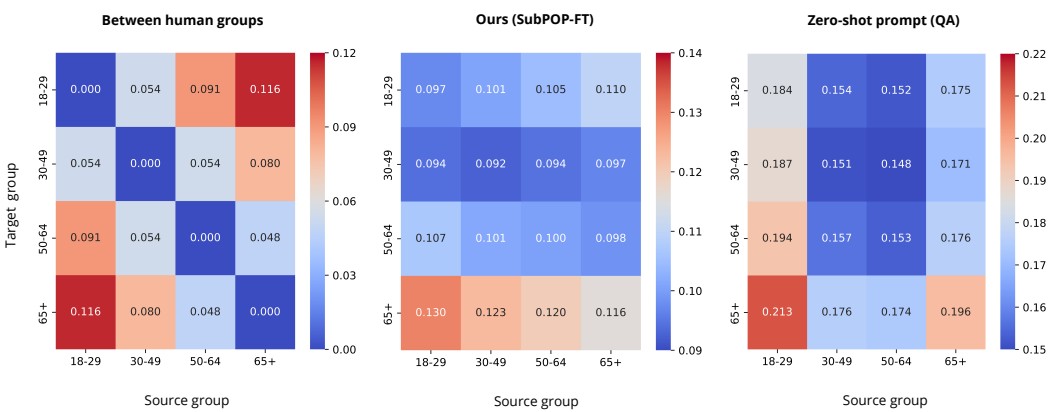

Figure 10: Heatmap of intergroup disagreement between a target human group (*y*-axis) and a source group (*x*-axis, either a human group or a group simulated with the language model), for OpinionQA evaluation data and age trait using Llama-2-7B as a base model. All subpopulations are unseen during fine-tuning.

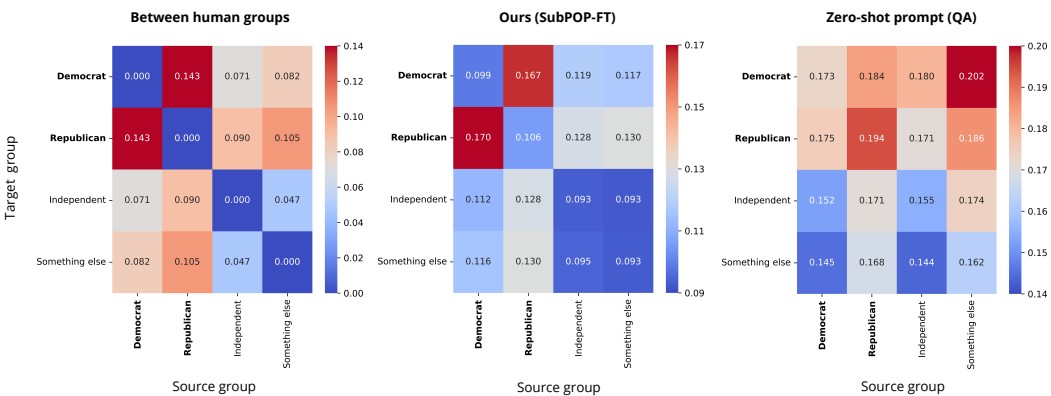

Figure 11: Heatmap of intergroup disagreement between a target human group (*y*-axis) and a source group (*x*-axis, either a human group or a group simulated with the language model), for OpinionQA evaluation data and political party (affiliation) trait using Llama-2-7B as a base model. Two subpopulations, Democrat and Republican, are seen during fine-tuning, while Independent and Something Else are unseen.

### F.4 Generalization to Unseen Subpopulations

Here we present a complete list of evaluation performance on OpinionQA for unseen subpopulations (the groups not used to fine-tune our model) and perform an analysis that shows our fine-tuned models are able to steer towards the given subpopulation information.

As shown in Table 7, we observe a performance improvement across unseen subpopulations. To verify that the performance improvements arise from the fine-tuned model being able to steer towards given subpopulations, we measure *intergroup disagreement* pattern for the demographic and ideology traits, shown in Figure 10, 11, 12, and 13. We consistently observe across traits that the disagreement pattern of our model resembles that of the human group, while zero-shot prompting with the base model exhibits a pattern completely different from the human group result. This observation shows that our fine-tuned model learns to condition on subpopulation information and also generalizes to subpopulations unseen during fine-tuning.

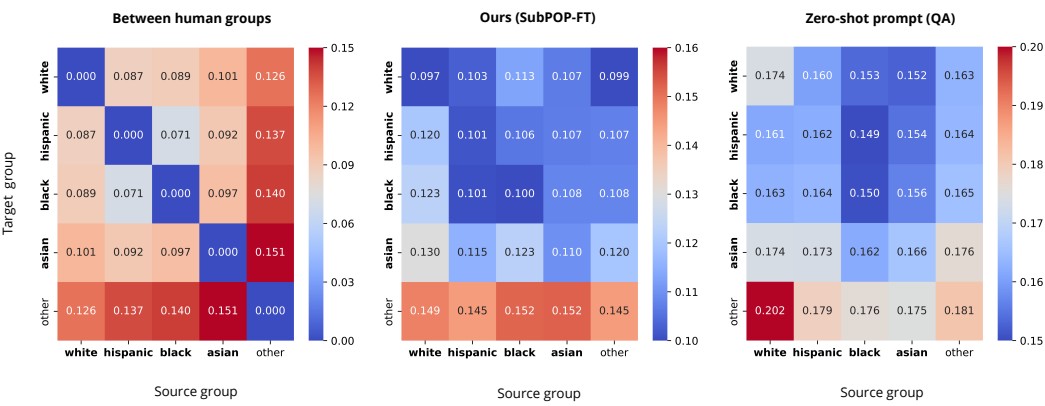

Figure 12: Heatmap of intergroup disagreement between a target human group ($y$-axis) and a source group ($x$-axis, either a human group or a group simulated with the language model), for OpinionQA evaluation data and race / ethnicity trait using Llama-2-7B as a base model. Four subpopulations except 'Other' are seen during fine-tuning. In this case, the model does not well predict the opinions of Other group. We suspect this occurs because Other is a group with highly diverse race or ethnicity backgrounds, making it inherently difficult to infer its opinion distribution from those of White, Hispanic, Black, and Asian subpopulations.

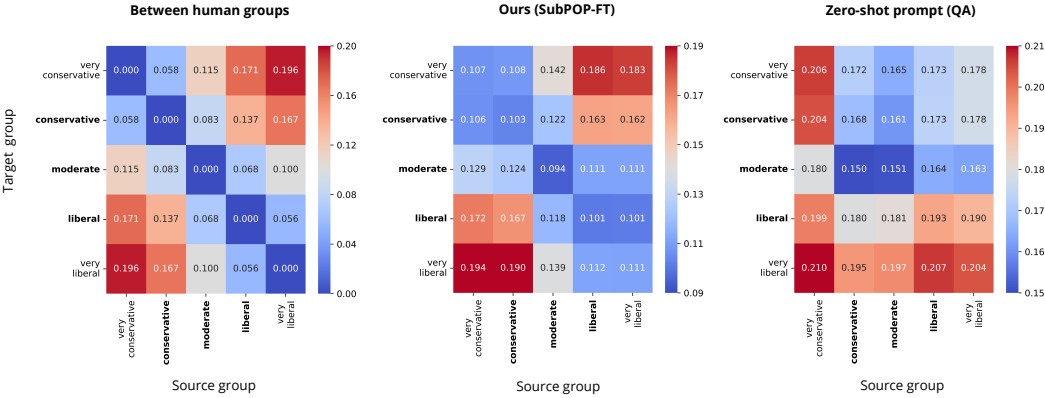

Figure 13: Heatmap of intergroup disagreement between a target human group ($y$-axis) and a source group ($x$-axis, either a human group or a group simulated with the language model), for OpinionQA evaluation data and political ideology trait using Llama-2-7B as a base model. Three subpopulations, Conservative, Moderate, and Liberal are seen during fine-tuning, while Very conservative and Very liberal are not seen.

## G Baseline Details

**Zero-shot prompting**: Three prompt styles—QA, BIO, and PORTRAY—are introduced in Santurkar et al. (2023) to integrate group information into prompts. These prompts are then combined with survey questions to construct inputs for LLM. Then, the first-token log-probability from LLM is measured to calculate the model's response distribution over options. In our baseline (and also in fine-tuning experiments) we focus on the QA steering format. Examples of this prompting method are shown in Figure 14.

**Few-shot prompting**: We craft a conditioning prompt that contains not only group information but also the group's response distribution to $k$ train questions, following Hwang et al. (2023). For a test question $q_{test} \in Q_{test}$, we first sort training questions $Q_{train}$ into $\{q_1, q_2, ...\}$ such that $\mathrm{sim}(\mathrm{E}(q_1), \mathrm{E}(q_{test})) > \mathrm{sim}(\mathrm{E}(q_2), \mathrm{E}(q_{test}))$, and so on. $\mathrm{E}(q)$ denotes the

embedding model (OpenAI-text-embedding-3-large) output of the input $q$ and sim is a cosine similarity between two embedding vectors. Then, response information of the first $k$ questions $\{q_i, p(\mathcal{A}_{q_i} | q_i, g)\}_{i=1}^{k}$ are used as few shot prompts to have the language model verbalize Meister et al. (2024) expected response distribution for the given $g$ and $q_{test}$. An example of the prompt for $k = 3$ case is shown in Figure 15, while we run the baseline experiment in a $k = 5$ setting.

**Modular Pluralism**: The intuition behind Modular Pluralism Feng et al. (2024) is that a language model trained on a text corpus of a specific subpopulation will faithfully represent public opinion of that population. Given a survey question with a PORTRAY-style steering prompt, each of language model 'modules' (fine-tuned Mistral-7B-Instruct-v0.1) generates an option choice with explanation. A black-box LLM (GPT-3.5-turbo-Instruct) receives all generations and select a generation that best aligns with the given group. Finally, using the chosen generation as a context, a black-box LLM generates probability distribution over options. The example pipeline is shown in Figure 16. Instead of the sub-sampled OpinionQA dataset the authors of the method used, we use the exactly same evaluation set across all baseline methods and our approach for a fair comparison.

**Upper bound**: We estimate the distribution between human responses and uniform distribution as an upper bound of WD metrics.

**Lower bound**: We compute a lower bound by randomly sampling a group of respondents and calculating the Wasserstein distance (WD) between the distribution of the sampled group and that of the original respondents for each question. We then bootstrap with $R = 1000$ to construct a 95% confidence interval (CI) for the WDs. Further details on this estimation process are provided below.

**Computing weighted answer distributions:** For each group $g$ and question $q$, we have $n_{gq}$ responses from respondents who belong to group $g$ answering question $q$: $x_1, x_2, \cdots, x_{n_{gq}}$, where $x_i \in \mathcal{A}_q$, i.e., the answer set for question $q$ (e.g., $\{1, 2, 3, 4\}$). Furthermore, each respondent (and thus, their response) is associated with a wave-specific weight $w_1, w_2, \cdots, w_{n_{gq}}$, provided by Pew Research. We compute the human answer distribution $\pi_{gq}^{(H)}$ as a weighted sum over responses, where the proportion of respondents providing answer $a \in \mathcal{A}_q$ is estimated as

$$\pi_{gq}^{(H)}(a) = \frac{\sum_{i=1}^{n_{gq}} w_i \mathbb{1}[x_i = a]}{\sum_{i=1}^{n_{gq}} w_i}.$$

**Bootstrapping at the respondent-level:** We draw bootstrap samples per group at the respondent-level including questions from all survey waves. This allows us to capture correlations in answer distributions across questions and across waves.

Specifically, let $\mathcal{P}_g$ represent the set of respondents in group $g$, where $|\mathcal{P}_g| = n_g$. We produce bootstrapped samples by repeatedly sampling $n_g$ respondents from $\mathcal{P}_g$ with replacement. Let $p_1^{(r)}, p_2^{(r)}, \cdots, p_{n_g}^{(r)}$ represent the sampled respondents for the $r$-th bootstrap, and let $w_1^{(r)}, w_2^{(r)}, \cdots, w_{n_g}^{(r)}$ represent their corresponding weights.

For each question $q$, let $\mathcal{P}_{gq} \subseteq \mathcal{P}_g$ represent the set of respondents from group $g$ who answered question $q$; as before, $|\mathcal{P}_{gq}| = n_{gq}$. Let us define $q(p_i)$ as person $p_i$'s response to question $q$ if $p_i$ answered question $q$, i.e., $p_i \in \mathcal{P}_{gq}$, and 0 otherwise. Then, we compute the $r$-th answer distribution to question $q$ as:

$$\pi_{gq}^{(r)}(a) = \frac{\sum_{i=1}^{n_g} \mathbb{1}[p_i^{(r)} \in \mathcal{P}_{gq}] w_i^{(r)} \mathbb{1}[q(p_i^{(r)}) = a]}{\sum_{i=1}^{n_g} \mathbb{1}[p_i^{(r)} \in \mathcal{P}_{gq}] w_i^{(r)}}.$$

**Human lower bound of WD.** Our statistic of interest is the mean Wasserstein distance over all questions $Q$ across all waves per group. We approximate this as the WD between

the observed human distribution $\pi_{gq}^{(H)}$ and the bootstrap sample $\pi_{gq}^{(r)}$ for question $q$ and group $g$. Over all $R = 1000$ bootstraps, we have

$$\mathcal{D}_g^{(H)} = \left\{ \frac{1}{|Q|} \sum_{q \in Q} WD(\pi_{gq}^{(H)}, \pi_{gq}^{(r)}) \right\}_{r=1}^R.$$

To quantify agreement between human samples, we report the mean and 95% CI (i.e., from $2.5^{th}$ to $97.5^{th}$ percentiles) of $\mathcal{D}_{gq}^{(H)}$.

## H Wave, Group-level Opinion Matching

Here we present a group-level and wave-level averaged Wasserstein distance. Wave-level result is in Table 8, and group-level results for OpinionQA and SubPOP-Eval are in Table 9, 10, respectively. We observe that the improvements in distribution matching between LLM response and human response are consistent across diverse subpopulations and waves.

Question: Which part of the United States do you currently live in?
A. Northeast
B. Midwest
C. South
D. West

Answer: A. Northeast

Answer the following question keeping in mind your previous answers.
Question: I am going to name some institutions in this country. As far as the people running these institutions are concerned, would you say you have a great deal of confidence, only some confidence, or hardly any confidence at all in them? Military
A. A great deal
B. Only some
C. Hardly any
D. Refused
Answer as a choice between A.,B.,C.,D.

Answer:

Question: What is your present religion, if any?
A. Protestant
B. Roman Catholic
C. Mormon
D. Orthodox
E. Jewish
F. Muslim
G. Buddhist
H. Hindu
I. Atheist
J. Agnostic
K. Other
L. Nothing in particular

Answer: I. Atheist

Answer the following question keeping in mind your previous answers.
Question: Do you believe there is a life after death?
A. Yes
B. No
C. Refused
Answer as a choice between A.,B.,C.

Answer:

Figure 14: Two examples of Zero-shot prompting in the QA format Santurkar et al. (2023). Subpopulation's information (colored in pink) is concatenated with survey question (colored in sky blue). The first-token log-probability (probabilities assigned to A, B, C, ...) are used to calculate language model's response distribution. The same steering prompt format is used in our fine-tuning experiment.

Table 4: American Trends Panel (ATP) wave topics for waves included in SubPOP-Train (top) and OpinionQA (bottom). Golden rows represent wave topics in SubPOP-Train that are not present in OpinionQA, and blue rows represent wave topics in OpinionQA that are not present in SubPOP-Train. For waves 68-79, survey questions related to COVID-19 (*e.g.*, contact tracing, vaccines, and relocation) were included as part of a survey along with the main survey topic.

| Wave | # questions | Wave Topic |
|---|---|---|
| 68 | 90 | American News Pathways, George Floyd, Black Lives Matter |
| 69 | 92 | Politics, 2020 Census |
| 70 | 56 | Religion in public life, social media's role in politics and society |
| 71 | 84 | Voter attitudes |
| 72 | 18 | New media |
| 73 | 82 | American News Pathways, social media |
| 74 | 51 | Online harassment, race relations |
| 75 | 18 | 2020 pre-election survey |
| 76 | 44 | American News Pathways |
| 77 | 13 | Culture of work |
| 78 | 57 | 2020 post-election survey |
| 79 | 93 | American News Pathways |
| 80 | 45 | Political priorities |
| 81 | 52 | Economics, pandemic financial outlook |
| 83 | 54 | Coronavirus vaccines and restrictions |
| 84 | 50 | Religion in politics and tolerance |
| 85 | 93 | News coverage of the Biden administration's first 100 days |
| 87 | 90 | Current political news and topics |
| 88 | 37 | Tech companies and policy issues |
| 90 | 79 | Twitter news attitudes |
| 91 | 64 | Benchmark study |
| 93 | 19 | Social media update |
| 95 | 78 | Politics timely and topical |
| 96 | 57 | Post-coronavirus pandemic spirituality |
| 98 | 76 | Coronavirus impacts on communities, living arrangements and life decisions |
| 99 | 20 | Artificial intelligence (AI) and human enhancement |
| 103 | 12 | Economic well-being |
| 104 | 92 | Politics, Religion in Public Life |
| 105 | 38 | Global Attitudes US Survey 2022 |
| 106 | 62 | Religion and the environment |
| 107 | 92 | Government and Parties |
| 108 | 83 | COVID and Climate, Energy and the Environment |
| 109 | 51 | New Digital Platforms and Gender Identity |
| 110 | 90 | Politics timely and topical |
| 111 | 23 | Online dating and E-commerce |
| 112 | 31 | Social media update |
| 113 | 53 | 2022 National Survey of Latinos (NSL) |
| 114 | 93 | Covid, scientists, and religion |
| 115 | 63 | Parents survey |
| 116 | 75 | Politics timely and topical |
| 117 | 16 | Religion and politics |
| 118 | 25 | Podcasts, news, and racial identity |
| 119 | 70 | AI and human enhancement |
| 120 | 61 | Politics timely and topical |
| 121 | 31 | Culture of work |
| 124 | 75 | Global Attitudes US Survey 2023 |
| 125 | 69 | Politics timely and topical |
| 126 | 93 | Racial attitudes, modern family |
| 127 | 59 | Americans and their data |
| 128 | 89 | Americans and planet Earth |
| 129 | 107 | Politics timely and topical |
| 130 | 94 | Politics representation |
| 131 | 70 | Gender and leadership |

| Wave | # questions | Wave Topic |
|---|---|---|
| 26 | 44 | Guns |
| 29 | 20 | Views on gender |
| 32 | 24 | Community types, Sexual harassment |
| 34 | 16 | Biomedical and food issues |
| 36 | 68 | Gender and leadership |
| 41 | 41 | Views of America in 2050 |
| 42 | 26 | Trust in science |
| 43 | 51 | Race in America |
| 45 | 13 | Misinformation |
| 49 | 19 | Privacy and surveillance |
| 50 | 43 | American families |
| 54 | 50 | Economic inequality |
| 82 | 56 | 2021 Global Attitudes Project U.S. survey |
| 92 | 23 | Political Typology |

Table 5: Comparison of evaluation performance for three response distribution modeling approaches, with Llama-2-7B as a base model. The last column (Explicit) is identical to the ours presented in Table 1. A model fine-tuned to predict the most probable choice (one-hot) performs the worst, as the model has not learned distributional opinion at fine-tuning phase. A model trained on augmented data (Aug. (×50, ×100)), while performing much better than one-hot still underperforms the explicit distribution modeling.

| Eval Dataset | One-hot | Aug. (× 50) | Aug. (× 100) | Explicit (Ours) |
|---|---|---|---|---|
| OpinionQA | 0.163 | 0.110 | 0.107 | 0.106 |
| SubPOP-Eval | 0.178 | 0.130 | 0.123 | 0.121 |

Table 6: Performance of the fine-tuned Llama-2-7B-chat model (Chat LLM FT). For comparison, we also present lower and upper bounds, the baseline method Zero-shot prompt (QA) and our fine-tuned Llama-2-7B (Base LLM FT).

| Method | OpinionQA | SubPOP-Eval |
|---|---|---|
| Upper bound (Unif.) | 0.178 | 0.208 |
| Lower bound (Human) | 0.031 | 0.033 |
| Base zero-shot prompt (QA) | 0.173 | 0.206 |
| Base LLM FT | 0.106 | 0.121 |
| Chat zero-shot prompt (QA) | 0.308 | 0.383 |
| Chat LLM FT | 0.109 | 0.148 |

Table 7: Evaluation performance of our fine-tuned Llama-2-7B model on OpinionQA for subpopulations not included in the fine-tuning dataset SubPOP-Train. For reference, we present a lower bound (human) and the zero-shot prompting (QA). Absolute difference refers to the WD difference between zero-shot prompting and ours, and the relative improvement is calculated in a same way as Figure 3. R. I. stands for Relative Improvement.

| Attribute | Group | Lower Bound | Zero-shot (QA) | Ours | Absolute Diff. | R. I. |
|---|---|---|---|---|---|---|
| Age | 18-29 | 0.023 | 0.185 | 0.096 | 0.089 | 0.548 |
| Age | 30-49 | 0.014 | 0.151 | 0.093 | 0.058 | 0.424 |
| Age | 50-64 | 0.014 | 0.154 | 0.101 | 0.052 | 0.377 |
| Age | 65+ | 0.013 | 0.195 | 0.115 | 0.080 | 0.438 |
| Region | Midwest | 0.016 | 0.153 | 0.095 | 0.058 | 0.425 |
| Region | West | 0.017 | 0.162 | 0.095 | 0.068 | 0.465 |
| Education | Associate's Degree | 0.026 | 0.159 | 0.098 | 0.061 | 0.455 |
| Education | High School Graduate | 0.017 | 0.144 | 0.092 | 0.053 | 0.413 |
| Education | Postgraduate | 0.015 | 0.174 | 0.106 | 0.068 | 0.426 |
| Education | Some College, No Degree | 0.018 | 0.144 | 0.093 | 0.051 | 0.405 |
| Income | $50,000-$75,000 | 0.016 | 0.153 | 0.098 | 0.054 | 0.396 |
| Income | $30,000-$50,000 | 0.019 | 0.144 | 0.094 | 0.050 | 0.400 |
| Political Ideology | Very Conservative | 0.026 | 0.208 | 0.107 | 0.101 | 0.555 |
| Political Ideology | Very Liberal | 0.025 | 0.202 | 0.111 | 0.091 | 0.514 |
| Political Party | Independent | 0.016 | 0.155 | 0.093 | 0.062 | 0.445 |
| Political Party | Something Else | 0.026 | 0.162 | 0.092 | 0.069 | 0.510 |
| Race | Other | 0.050 | 0.180 | 0.144 | 0.036 | 0.275 |
| Religion | Agnostic | 0.028 | 0.189 | 0.115 | 0.074 | 0.459 |
| Religion | Buddhist | 0.063 | 0.207 | 0.149 | 0.059 | 0.405 |
| Religion | Nothing in Particular | 0.019 | 0.153 | 0.092 | 0.061 | 0.454 |
| Religion | Orthodox | 0.083 | 0.221 | 0.180 | 0.041 | 0.298 |
| Religion | Other | 0.051 | 0.184 | 0.123 | 0.061 | 0.457 |
| Religion | Roman Catholic | 0.018 | 0.145 | 0.098 | 0.047 | 0.371 |

Predict the answer distribution of the following questions for people whose **race or ethnicity** is **White**.

Question: Overall, how does being Asian affect people's ability to get ahead in our country these days?
A. Helps a lot
B. Helps a little
C. Hurts a little
D. Hurts a lot
E. Neither helps nor hurts
F. Refused

Answer distribution: {'A': 0.050, 'B': 0.210, 'C': 0.241, 'D': 0.034, 'E': 0.474, 'F': 0.010}

Question: Thinking about the people who design artificial intelligence computer programs, how well do you think they take into account the experiences and views of Asian adults?
A. Very well
B. Somewhat well
C. Not too well
D. Not at all well
E. Not sure
F. Refused

Answer distribution: {'A': 0.078, 'B': 0.216, 'C': 0.138, 'D': 0.069, 'E': 0.507, 'F': 0.008}

Question: Thinking about the people who design artificial intelligence computer programs, how well do you think they take into account the experiences and views of Hispanic adults?
A. Very well
B. Somewhat well
C. Not too well
D. Not at all well
E. Not sure
F. Refused

Answer distribution: {'A': 0.050, 'B': 0.160, 'C': 0.203, 'D': 0.090, 'E': 0.505, 'F': 0.008}

Question: Thinking again about race and race relations in the U.S. in general, how well, if at all, do you think each of these groups get along with each other in our society these days? Hispanics and Asians
A. Very well
B. Pretty well
C. Not too well
D. Not at all well
E. Not sure
F. Refused

Answer distribution:

Figure 15: Few-shot prompting example for $k = 3$. Group information is presented in the beginning of the prompt (colored in pink). Following group information, $k$ questions whose text embedding are the most similar to the text embedding of the evaluation question (colored in sky blue) are presented along with their opinion distribution. $k$ questions are presented in the ascending order of cosine similarity. The generation of language model (verbalization of opinion distribution) is parsed to obtain the response distribution.

In terms of **political party**, you are **Republican**. Please respond to the following question with explanation.

Question: Please indicate whether the following is a major reason, a minor reason, or not a reason why you own a gun. As part of a gun collection
A. Major reason
B. Minor reason
C. Not a reason

Answer:

---

Which of the following comments best reflect the people of **Republican** in terms of **political party**?

Comment 1: **(generation from model 1)**

Comment 2: **(generation from model 2)**

Comment 3: **(generation from model 3)**

Comment 4: **(generation from model 4)**

Comment 5: **(generation from model 5)**

Comment 6: **(generation from model 6)**

Please select one comment number from 1 to 6.

---

In terms of **political party**, you are **Republican**. Please respond to the following question with the help of a passage.

Passage: **(selected generation)**

Question: Please indicate whether the following is a major reason, a minor reason, or not a reason why you own a gun. As part of a gun collection
A. Major reason
B. Minor reason
C. Not a reason

Answer:

Figure 16: Pipeline example of Modular Pluralism. Given a demographic group and a survey question, the first prompt is asked to multiple (6) language models, Mistral-7B-v0.1-Instruct fine-tuned on the community text corpus. The generations are sent to a black-box LLM (gpt-3.5-0613-Instruct) in the format of the second prompt. The black-box LLM answers which one of generations best reflects the given demographics. Finally, the selected generation serves as a context to answer the given survey question and the black-box LLM is prompted (the third prompt) to generate response distribution over the answer token A, B, C, etc.

Table 8: Per-wave Wasserstein distance on OpinionQA for each base model, comparing baseline zero-shot prompting (QA) with our fine-tuned model Ours(SubPOP-FT). Highlighted rows represent waves whose topics are not covered by the training data (SubPOP-Train). We observe WD improvement consistently across survey waves and also for waves of topics not covered in the training data.

| Wave | Llama-2-7B Zero-shot | Llama-2-7B Ours | Llama-2-13B Zero-shot | Llama-2-13B Ours | Mistral-7B-v0.1 Zero-shot | Mistral-7B-v0.1 Ours | Llama-3-70B Zero-shot | Llama-3-70B Ours |
|---|---|---|---|---|---|---|---|---|
| 26 | 0.191 | 0.145 | 0.180 | 0.126 | 0.178 | 0.131 | 0.134 | 0.084 |
| 29 | 0.169 | 0.096 | 0.172 | 0.123 | 0.153 | 0.096 | 0.125 | 0.085 |
| 32 | 0.163 | 0.110 | 0.156 | 0.098 | 0.137 | 0.099 | 0.151 | 0.091 |
| 34 | 0.155 | 0.105 | 0.171 | 0.089 | 0.134 | 0.095 | 0.138 | 0.083 |
| 36 | 0.175 | 0.120 | 0.184 | 0.126 | 0.175 | 0.107 | 0.130 | 0.087 |
| 41 | 0.160 | 0.090 | 0.155 | 0.084 | 0.134 | 0.073 | 0.116 | 0.085 |
| 42 | 0.159 | 0.053 | 0.146 | 0.059 | 0.127 | 0.059 | 0.131 | 0.084 |
| 43 | 0.179 | 0.112 | 0.172 | 0.104 | 0.154 | 0.102 | 0.124 | 0.099 |
| 45 | 0.177 | 0.101 | 0.177 | 0.093 | 0.149 | 0.084 | 0.126 | 0.091 |
| 49 | 0.151 | 0.098 | 0.143 | 0.131 | 0.128 | 0.116 | 0.159 | 0.087 |
| 50 | 0.209 | 0.139 | 0.196 | 0.121 | 0.188 | 0.125 | 0.154 | 0.078 |
| 54 | 0.158 | 0.087 | 0.158 | 0.087 | 0.128 | 0.077 | 0.118 | 0.079 |
| 82 | 0.173 | 0.098 | 0.171 | 0.075 | 0.148 | 0.077 | 0.174 | 0.093 |
| 92 | 0.165 | 0.073 | 0.153 | 0.071 | 0.140 | 0.055 | 0.126 | 0.081 |

Table 9: Per-group Wasserstein distance on OpinionQA for each base models, before and after fine-tuning on SubPOP-Train. Base refers to zero-shot prompting (QA) and FT refers to Fine-tuned model performance. *Full group variable name is "College grad, some Postgrad".

| Attribute | Group | Human Baseline | Llama-2-7B Base | Llama-2-7B FT | Llama-2-13B Base | Llama-2-13B FT | Mistral-7B-v0.1 Base | Mistral-7B-v0.1 FT | Llama-3-70B Base | Llama-3-70B FT |
|---|---|---|---|---|---|---|---|---|---|---|
| Region | Northeast | 0.023 | 0.165 | 0.094 | 0.155 | 0.088 | 0.155 | 0.083 | 0.134 | 0.084 |
| | South | 0.017 | 0.149 | 0.092 | 0.143 | 0.085 | 0.133 | 0.081 | 0.113 | 0.078 |
| Education | College grad* | 0.018 | 0.165 | 0.099 | 0.157 | 0.096 | 0.136 | 0.089 | 0.125 | 0.085 |
| | Less than high school | 0.043 | 0.161 | 0.101 | 0.150 | 0.096 | 0.134 | 0.094 | 0.151 | 0.091 |
| Gender | Male | 0.015 | 0.182 | 0.093 | 0.152 | 0.089 | 0.131 | 0.083 | 0.138 | 0.083 |
| | Female | 0.013 | 0.162 | 0.100 | 0.158 | 0.092 | 0.146 | 0.088 | 0.130 | 0.087 |
| Race / ethnicity | Black | 0.031 | 0.151 | 0.102 | 0.144 | 0.095 | 0.132 | 0.091 | 0.116 | 0.085 |
| | White | 0.012 | 0.176 | 0.097 | 0.178 | 0.093 | 0.145 | 0.085 | 0.131 | 0.084 |
| | Asian | 0.051 | 0.165 | 0.111 | 0.167 | 0.104 | 0.143 | 0.102 | 0.124 | 0.099 |
| | Hispanic | 0.044 | 0.162 | 0.102 | 0.163 | 0.098 | 0.134 | 0.092 | 0.126 | 0.091 |
| Income | $100,000 or more | 0.019 | 0.172 | 0.103 | 0.162 | 0.100 | 0.147 | 0.091 | 0.159 | 0.087 |
| | Less than $30,000 | 0.021 | 0.162 | 0.091 | 0.148 | 0.083 | 0.127 | 0.080 | 0.154 | 0.078 |
| Political Party | Democrat | 0.016 | 0.172 | 0.099 | 0.158 | 0.092 | 0.161 | 0.082 | 0.118 | 0.079 |
| | Republican | 0.019 | 0.196 | 0.105 | 0.235 | 0.101 | 0.181 | 0.095 | 0.174 | 0.093 |
| Political Ideology | Liberal | 0.022 | 0.192 | 0.100 | 0.181 | 0.094 | 0.166 | 0.084 | 0.126 | 0.081 |
| | Conservative | 0.021 | 0.169 | 0.103 | 0.153 | 0.099 | 0.144 | 0.094 | 0.141 | 0.092 |
| | Moderate | 0.016 | 0.151 | 0.094 | 0.153 | 0.090 | 0.132 | 0.082 | 0.106 | 0.081 |
| Religion | Protestant | 0.016 | 0.015 | 0.166 | 0.096 | 0.158 | 0.092 | 0.146 | 0.086 | 0.143 |
| | Jewish | 0.058 | 0.182 | 0.124 | 0.182 | 0.165 | 0.115 | | 0.144 | 0.115 |
| | Hindu | 0.079 | 0.211 | 0.160 | 0.232 | 0.163 | 0.211 | 0.161 | 0.181 | 0.157 |
| | Atheist | 0.035 | 0.202 | 0.118 | 0.204 | 0.110 | 0.196 | 0.099 | 0.135 | 0.098 |
| | Muslim | 0.089 | 0.202 | 0.159 | 0.209 | 0.156 | 0.204 | 0.146 | 0.171 | 0.144 |

Table 10: Per-group Wasserstein distance on SubPOP-Eval for each base models, before and after fine-tuning on SubPOP-Train. Base refers to zero-shot prompting (QA). *Full group variable name is "College grad, some Postgrad".

| Attribute | Group | Human Baseline | Llama-2-7B Base | FT | Llama-2-13B Base | FT | Mistral-7B-v0.1 Base | FT | Llama-3-70B Base | FT |
|---|---|---|---|---|---|---|---|---|---|---|
| Region | Northeast | 0.027 | 0.196 | 0.113 | 0.193 | 0.103 | 0.185 | 0.108 | 0.156 | 0.078 |
| | South | 0.018 | 0.183 | 0.108 | 0.185 | 0.103 | 0.176 | 0.103 | 0.138 | 0.080 |
| Education | College grad* | 0.019 | 0.206 | 0.105 | 0.175 | 0.101 | 0.167 | 0.099 | 0.137 | 0.077 |
| | Less than high school | 0.036 | 0.191 | 0.129 | 0.182 | 0.117 | 0.172 | 0.121 | 0.180 | 0.108 |
| Gender | Male | 0.017 | 0.186 | 0.102 | 0.176 | 0.101 | 0.170 | 0.099 | 0.150 | 0.079 |
| | Female | 0.016 | 0.184 | 0.108 | 0.198 | 0.105 | 0.176 | 0.100 | 0.151 | 0.080 |
| Race / ethnicity | Black | 0.029 | 0.200 | 0.114 | 0.179 | 0.102 | 0.170 | 0.107 | 0.139 | 0.094 |
| | White | 0.014 | 0.190 | 0.105 | 0.187 | 0.103 | 0.181 | 0.102 | 0.153 | 0.083 |
| | Asian | 0.049 | 0.201 | 0.119 | 0.190 | 0.107 | 0.184 | 0.114 | 0.158 | 0.096 |
| | Hispanic | 0.050 | 0.204 | 0.133 | 0.199 | 0.122 | 0.182 | 0.134 | 0.172 | 0.115 |
| Income | $100,000 or more | 0.021 | 0.210 | 0.111 | 0.184 | 0.106 | 0.176 | 0.102 | 0.179 | 0.082 |
| | Less than $30,000 | 0.026 | 0.179 | 0.115 | 0.172 | 0.103 | 0.165 | 0.105 | 0.171 | 0.086 |
| Political Party | Democrat | 0.020 | 0.219 | 0.103 | 0.197 | 0.092 | 0.199 | 0.091 | 0.128 | 0.076 |
| | Republican | 0.023 | 0.205 | 0.123 | 0.234 | 0.117 | 0.206 | 0.115 | 0.187 | 0.093 |
| Political Ideology | Liberal | 0.019 | 0.224 | 0.102 | 0.191 | 0.090 | 0.188 | 0.096 | 0.134 | 0.076 |
| | Conservative | 0.022 | 0.184 | 0.120 | 0.178 | 0.112 | 0.172 | 0.113 | 0.160 | 0.092 |
| | Moderate | 0.018 | 0.191 | 0.110 | 0.183 | 0.103 | 0.170 | 0.103 | 0.141 | 0.082 |
| Religion | Protestant | 0.019 | 0.187 | 0.110 | 0.179 | 0.107 | 0.172 | 0.105 | 0.164 | 0.082 |
| | Jewish | 0.066 | 0.245 | 0.149 | 0.226 | 0.144 | 0.218 | 0.129 | 0.164 | 0.119 |
| | Hindu | 0.095 | 0.264 | 0.180 | 0.253 | 0.169 | 0.252 | 0.186 | 0.223 | 0.166 |
| | Atheist | 0.021 | 0.222 | 0.126 | 0.207 | 0.103 | 0.199 | 0.116 | 0.132 | 0.106 |
| | Muslim | 0.090 | 0.253 | 0.175 | 0.240 | 0.181 | 0.238 | 0.173 | 0.203 | 0.158 |

