# OpenReview forum: "Language Model Fine-Tuning on Scaled Survey Data for Predicting Distributions of Public Opinions"
_colmweb.org/COLM/2025/Workshop/Social_Sim — Social Sim'25_

### Official Review · Reviewer_cGvE · 2025-07-15
**Review of Language Model Fine-Tuning on Scaled Survey Data for Predicting Distributions of Public Opinions**

**Rating:** 6
**Overall Assessment:** 3
**Confidence:** 5

**Review:**

The paper is well written and clearly explains how the study was conducted. I appreciate the experimental setup, particularly the use of upper and lower bounds, baselines, and the choice of evaluation metric to assess the alignment between model and subpopulation response distributions. The intergroup disagreement figures also provide valuable insight into the variance within subgroups. While the finding that fine-tuning enables models to better mimic human responses may not be highly surprising, it is nonetheless a meaningful contribution, especially when validated on a large-scale public opinion dataset amid growing interest in this area. I would be particularly interested in further analysis of intersectional demographic groups and how the model performs across these more complex subpopulations.

**Comments Suggestions And Typos:**

P.21 line 749 : "... while the baseline method performs worse then ... " should be "... while the baseline method performs worse than ... "

**Paper Summary:**

This paper proposes fine-tuning large language models (LLMs) on large-scale, high-quality survey data consisting of questions and responses from diverse demographic, socioeconomic, and ideological subpopulations. The performance of the fine-tuned LLMs is evaluated using the Wasserstein distance, which measures how closely the model’s output distribution aligns with actual human survey responses distributions. The study finds that fine-tuned LLMs reduce this distance by 32–46% compared to top-performing baselines. The author(s) also mention that the models also generalize well to out-of-distribution data, including previously unseen subpopulations, new survey topics, and different survey families. Additionally, the author(s) curate and release a dataset of 70,000 subpopulation-response distribution pairs called SubPOP.

**Relevance:**

4

**Summary Of Strengths:**

I think the main strength of this paper lies in its effort to evaluate generalization to unseen subpopulations. The results also reinforce existing research suggesting that fine-tuning can reduce discrepancies between how humans actually respond, write, or behave and how models perform. Additionally, the release of a large-scale dataset containing subpopulation–response distribution pairs is a valuable contribution that can support further research in this area.

**Summary Of Weaknesses:**

While I appreciate the use of intergroup disagreement as an evaluation approach, I found the interpretation of Figure 4 somewhat unclear, as the caption could be more precise.

1. Among the various demographic attributes considered, the paper highlights education level in the main text. Based on the figure, it appears that groups with similar education levels exhibit strong agreement in their responses, whereas groups with differing education levels show greater disagreement. This suggests that education level may be a strong predictor of response patterns in public opinion data. I would have liked to see more discussion around this implication. Similarly, age (Figure 10) and political ideology (Figure 13) also appear to correlate strongly with response distributions, which could be explored further.

2. Given that the authors chose specific subgroups for fine-tuning, I was unclear whether the observed improvements in the heatmap are primarily driven by those subgroup choices or whether the fine-tuning leads to general improvement regardless of which groups are used. In other words, would similar gains be observed if we fine-tuned on a small, representative sample across all groups rather than targeted subpopulations?

3. This suggestion may be a bit of a stretch, but it could be interesting to analyze whether certain survey waves are more strongly associated with particular demographic groups, and whether these associations persist post fine-tuning. Along similar lines, the conclusion could be strengthened by going beyond the general claim that "fine-tuning improves human alignment" and instead offering more specific insights into how fine-tuning affects model behavior across different demographic or topical dimensions.

---

### Official Review · Reviewer_dw1s · 2025-07-18
**Language Model Fine-Tuning on Scaled Survey Data for Predicting Distributions of Public Opinions**

**Rating:** 6
**Overall Assessment:** 2
**Confidence:** 3

**Review:**

This paper submission provides novel method and dataset construction for advancing opinion prediction, but its geographic/demographic scope as well as potential LLM bias may encompass limitations towards broader real-world applicability. Future improvements can consider: 1) expandingSubPOP with global surveys (e.g., WVS);  2) integrating parameter-efficient fine-tuning;  and 3) more rigorously auditing cultural and response biases.

**Comments Suggestions And Typos:**

Make Figure 5, 14, etc. wider (more effective usage of space)

**Paper Summary:**

This paper focuses on the novel angle of fine-tuning LLMs to predict response distributions in the task domain of survey response prediction (relevant for early stages of survey design) through leveraging unique structural characteristics of survey data. Experimental results showcase that the proposed method of finetuning on curated SubPOP dataset reduces LLM-human gap by up to 46% compared to baselines, thereby enabling more efficient survey designs.

**Relevance:**

4

**Summary Of Strengths:**

Proposed LLM training methodology leverages structural advantages of survey data, such as explicit subpopulation-opinion links and/or calibrated representativeness, that are rarely found in pretraining corpora.

The constructed data to support model training encompasses 70K subpopulation-response pairs (3,362 questions) from authoritative surveys (Pew ATP, GSS), 6.5× larger than predecessors like OpinionQA. Serves as a potentially useful resource to our community.

**Summary Of Weaknesses:**

The dataset construction contribution, SubPOP, focuses on US-centric surveys (ATP, GSS), while overlooking underrepresentation of non-Western, non-English-speaking, or intersectional subgroups, which may perpetuate biases demonstrated in prior work. Further validation of cultural/linguistic generalization would be good.

The current evaluation scope of LLM survey simulations involves distributional metrics (Wasserstein distance) but overlooks response consistency biases (e.g., ordering effects, option labeling) noted in LLM survey simulations. Testing on more complex, open-ended questions where LLMs exhibit thematic biases would also be interesting.

Discussion of temporal dynamics over longitudinal studies can also bridge theoretical gaps in this computational social science subarea.

---

### Meta-Review · Program_Chairs · 2025-07-24

**Recommendation:** Accept

**Metareview:**

Please incorporate the feedback!